# High-Frequency Transcranial Random Noise Stimulation over the Left Prefrontal Cortex Increases Resting-State EEG Frontal Alpha Asymmetry in Patients with Schizophrenia

**DOI:** 10.3390/jpm12101667

**Published:** 2022-10-07

**Authors:** Ta-Chuan Yeh, Cathy Chia-Yu Huang, Yong-An Chung, Jooyeon Jamie Im, Yen-Yue Lin, Chin-Chao Ma, Nian-Sheng Tzeng, Chuan-Chia Chang, Hsin-An Chang

**Affiliations:** 1Department of Psychiatry, Tri-Service General Hospital, National Defense Medical Center, Taipei 114201, Taiwan; 2Department of Life Sciences, National Central University, Taoyuan 320317, Taiwan; 3Department of Nuclear Medicine, College of Medicine, The Catholic University of Korea, Seoul 21431, Korea; 4Department of Emergency Medicine, Tri-Service General Hospital, National Defense Medical Center, Taipei 114202, Taiwan; 5Department of Emergency Medicine, Taoyuan Armed Forces General Hospital, Taoyuan 325208, Taiwan; 6Department of Psychiatry, Tri-Service General Hospital Beitou Branch, National Defense Medical Center, Taipei 112003, Taiwan

**Keywords:** transcranial random noise stimulation, electroencephalography, frontal alpha asymmetry, schizophrenia, negative symptoms

## Abstract

Reduced left-lateralized electroencephalographic (EEG) frontal alpha asymmetry (FAA), a biomarker for the imbalance of interhemispheric frontal activity and motivational disturbances, represents a neuropathological attribute of negative symptoms of schizophrenia. Unidirectional high-frequency transcranial random noise stimulation (hf-tRNS) can increase the excitability of the cortex beneath the stimulating electrode. Yet, it is unclear if hf-tRNS can modulate electroencephalographic FAA in patients with schizophrenia. We performed a randomized, double-blind, sham-controlled clinical trial to contrast hf-tRNS and sham stimulation for treating negative symptoms in 35 schizophrenia patients. We used electroencephalography to investigate if 10 sessions of hf-tRNS delivered twice-a-day for five consecutive weekdays would modulate electroencephalographic FAA in schizophrenia. EEG data were collected and FAA was expressed as the differences between common-log-transformed absolute power values of frontal right and left hemisphere electrodes in the alpha frequency range (8–12.5 Hz). We found that hf-tRNS significantly increased FAA during the first session of stimulation (*p* = 0.009) and at the 1-week follow-up (*p* = 0.004) relative to sham stimulation. However, FAA failed to predict and surrogate the improvement in the severity of negative symptoms with hf-tRNS intervention. Together, our findings suggest that modulating electroencephalographic frontal alpha asymmetry by using unidirectional hf-tRNS may play a key role in reducing negative symptoms in patients with schizophrenia.

## 1. Introduction

Enduring and primary negative symptoms (i.e., avolition, alogia, blunted affect, anhedonia, and asociality) are core components of schizophrenia strongly associated with long-term morbidity and poor psychosocial and occupational functioning [1] that substantially contribute to the burden of disease [2]. However, these symptoms are considered only marginally responsive to pharmacological treatment and patients with these symptoms account for a noticeable subpopulation of schizophrenia, suggesting an urgent need for novel and effective treatments for negative symptoms.

In the neurodevelopmental hypothesis, a failure in the process of normal brain lateralization, abnormal structural brain asymmetry, and asymmetry of functional connectivity plays a vital role in the pathophysiology of schizophrenia [3]. The balance of interhemispheric activity is crucial for maintaining mental health and hemispheric lateralization of brain activity may represent a neurophysiological endophenotype of schizophrenia [3]. For example, research indicated that patients with schizophrenia had more global gray matter asymmetry than control, and increased gray matter asymmetry was associated with negative symptoms, e.g., avolition [4].

Research indicates that structural, functional, and metabolic abnormalities are dominant in the mid-frontal region of patients with schizophrenia [5]. Electroencephalographic (EEG) frontal alpha asymmetry (FAA), particularly in the midfrontal area, has been suggested to be a biomarker for the imbalance of interhemispheric activity of schizophrenia [6,7]. FAA is expressed as the difference between absolute power values of frontal right and left hemisphere electrodes in the alpha frequency range. Since EEG alpha power is assumed to be inversely related to cortical activation [8], reduced left frontal alpha power relative to right frontal alpha power reflects an increase in left frontal activity [9]. Earlier research reported that patients with schizophrenia had significantly greater left lateralized alpha power than controls, indicating a deficit in resting-state left frontal activity, which may reflect diminished left lateralization related to “disconnections” across wider fronto-temporal networks [10]. Motivational impairment is arguably the single most important factor that contributes to negative symptoms of schizophrenia, in particular, the ‘avolition-apathy’ subdomain. In the motivational direction model of asymmetric frontal brain activity [11], there are two proposed distinct and fundamental motivational systems including one supporting approach behaviors and the other supporting withdrawal behaviors. The approach and withdrawal system are associated with separate neural circuits involving different frontal regions. Specifically, increases in left and right frontal brain activation are associated with approach and withdrawal motivation, respectively. In schizophrenia, the overall pattern of motivational disturbance is characterized by both diminished approach and elevated withdrawal dispositions. Research applying FAA as a tool to examine the motivational direction model of asymmetric frontal brain activity in schizophrenia indicates a reduced left-lateralized FAA in patients compared to controls. This suggests that left-lateralized FAA may underlie diminished approach motivation and represent a neuropathological attribute of these patients [6,7].

Recent research has suggested that sub-threshold noninvasive brain stimulation (NIBS) interventions may be effective in treating negative symptoms [12,13]. Our pilot trial evaluating unidirectional high-frequency transcranial random noise stimulation (hf-tRNS) over the left prefrontal cortex for the treatment of negative symptoms of schizophrenia achieved the primary efficacy endpoint [14]. tRNS delivers randomly alternating current that follows a white noise structure (i.e., all frequencies from 0.1 Hz to 640 Hz, with the same power and a Gaussian amplitude structure). High-frequency (101–640 Hz) tRNS has been reported to increase cortical excitability of the primary motor cortex with sustained after-effects [15,16], possibly through the repetitive opening of the voltage-gated Na^+^ channels that shortens the hyperpolarization phase [15] and through stochastic resonance (SR) phenomenon that modulates cortical signal-to-noise ratio [17]. Given that unidirectional hf-tRNS potentially increases excitability in the cortex under the anode (i.e., left hemispheric lateral prefrontal) and thereby restores the imbalance of interhemispheric frontal activity of schizophrenia, the present study aimed to further explore the effects of unidirectional hf-tRNS targeting left prefrontal cortex on electroencephalographic FAA in the mid-frontal region. We hypothesized that hf-tRNS reduced the negative symptom severity of the participants by increasing their FAA in the midfrontal area. Exploratory analyses were also performed to assess the predictive and surrogate characteristics of FAA as biomarkers for the treatment response to aid patient selection for this promising intervention.

## 2. Materials and Methods

### 2.1. Participants and Study Design

The participants who met DSM-5 defined schizophrenia or schizoaffective disorder and were symptomatically stable on antipsychotic treatments (Table 1) were recruited in the trial (ClinicalTrials.gov, NCT04038788) that was approved by the local ethics committee and carried out at the Tri-Service General Hospital, Taipei, Taiwan. In this randomized, double-blind, and sham-controlled clinical trial, participants were randomly and equally assigned to either hf-tRNS or sham condition. The blinding integrity was achieved by a study coordinator who was not involved in the patient care or contact and created 5-digit random numbers and assigned the randomization numbers to the participants. The negative symptom severity as the primary outcome was rated at baseline, the end of stimulation, 1-week and 1-month follow-up visits by using the Positive and Negative Syndrome Scale Factor Score for Negative Symptoms (PANSS-FSNS) consisting of items N1-4, N6, G7, and G16 of the PANSS [18]. Two subdomains of negative symptoms were identified from the PANSS: the Expressive Deficit domain (EXP domain, the sum of items of N1, N3, N6, G5, G7, and G13) including features of blunted affect, alogia, and the Avolition-Apathy domain (AA domain, the sum of items of N2, N4, and G16) including manifestations of anhedonia, asociality, and avolition. The study also obtained other PANSS factor scores for positive symptoms (PANSS-FSPS), excitement, disorganization, and emotional distress [19,20], and the total score of the Extrapyramidal Symptoms Rating Scale (ESRS), a quantitative measure of antipsychotic-induced extrapyramidal movement symptoms [21]. The ESRS is composed of the patient-assessed ESRS questionnaire and the physician-rated subscales for assessing parkinsonism/akathisia, dystonia movement, and dyskinesia movement. The clinical results and detailed information regarding the inclusion and exclusion criteria were reported elsewhere [14]. Here, EEG data from 35 participants were analyzed. Some of the methods can be found in our previous publications [14,22] but the procedures are outlined here for completeness (Please also see Appendix A for details of the time points where the assessments are conducted).

### 2.2. Brain Stimulation

A battery-operated device (Eldith Stimulator Plus, NeuroConn, Ilmenau, Germany) connected with an equalizer extension box [23] for the high-density 4 × 1 electrode montage [24] was used for stimulation. Five carbon rubber electrodes (radius = 1 cm, area = 3 cm^2^, thickness = 1 mm) with an anode placed over AF3 of the international 10-10 EEG system and cathodes over AF4, F2, F6, and FC4 were applied to the scalp with conductive paste and were checked for the combined impedance of all electrodes < 15 kΩ. The electrode montage applied in the current study was adapted from previous tRNS research for stimulation over the left prefrontal cortex and for further EEG measurement during tRNS [25]. In the active hf-tRNS group, 10 sessions of 20-min, and 2 mA-intensity random noise stimulation with 100–640 Hz frequency, 1 mA offset, and 15 s ramp-in/ramp-out were delivered twice a day for five consecutive weekdays. The 1 mA offset was to make sure that the current flow was unidirectional, i.e., delivering current flow analogously to transcranial direct current stimulation through switching the offset from zero to 1 mA to prevent the oscillations from being negatively polarized(Figure 1). Sham stimulation delivered 40-sec, 2 mA normal-like stimulation, followed by a tiny current pulse (110 μA over 15 ms) for impedance control taking place every 550 ms for the remaining time. The participants sat comfortably and kept their eyes open during stimulation unless otherwise specified. The break between twice-daily stimulation sessions was at least 2 h.

### 2.3. EEG at Rest and during Stimulation

Participants sat in a recliner in a sound-attenuated room. Alcohol and caffeinated beverages were prohibited before EEG recording to minimize potential confounders. EEG data at rest were recorded with eyes closed (5 min) and eyes open (5 min) at baseline, the end of stimulation, and the one-week follow-up using a 32-channel EEG cap (NP32, GmbH, Ilmenau, Germany) with the international 10–20 system Ag/AgCl sintered ring electrodes together with Neuro Prax^®^ TMS/tES compatible full band DC-EEG system (NeuroConn GmbH, Ilmenau, Germany). The order of eyes-open/eyes-closed conditions was randomized and counterbalanced across participants. The impedance of each electrode was checked to achieve below 5 kΩ. The sampling frequency was set at 4000 Hz with an analogous bandpass filter (0–1200 Hz) and an analog-digital precision of 24 bits. The reference electrode was placed at the tip of the nose and the ground electrode was placed at Fpz. Eye blinks and vertical electrooculogram (VEOG) were recorded by two electrodes placed above and below the left eye, respectively. A horizontal electrooculogram (HEOG) was recorded by two electrodes placed at 1 cm from the outer canthi of both eyes. The participants underwent 3-min calibration tasks before EEG recording to assess the impact of the blink artifacts and vertical/horizontal movements. These artifacts were automatically removed by a real-time method built in the Neuro Prax^®^ EEG system software.

EEG data during the first session of stimulation were recorded by using the Neuro Prax^®^ TMS/tES system with a DC-coupled EEG amplifier. The application of NeuroPrax’s built-in algorithm with similar recording parameters, closed-loop stimulation protocol, and online correction method has been shown to effectively capture and remove the artifacts induced by transcranial electrical stimulation (tES) [26,27]. Briefly, analog signals of hf-tRNS were derived from the Eldith Stimulator Plus as reference signals. The galvanic isolated signals (amplitude +/−40 mV) were fed into the EEG amplifier together with EEG, EOG, and ECG signals. The galvanic isolation of the reference signals ensures the electrical safety of the patients. A short-time learning period (around 10 s) was used to calculate the scaling factor of the regression model. Meanwhile, a custom-made masking device was applied to avoid breaching the blindness by active stimulation-induced distortion of the EEG signals shown on the screen. Next, the tES-induced signal parts during the whole stimulation period of 20 min were eliminated by the online correction procedure. tRNS-EEG data were recorded for 20 min with eyes open.

Offline, the EEG data were down-sampled to 500 Hz, band-pass filtered to 1–100 Hz, and analog 60 Hz-notch filtered by using EEGLAB v2020.0, an open-source toolbox for signal processing [28]. The artifact subspace reconstruction (ASR) method was applied to automatically detect and remove the bad channels [29]. Independent component analysis (ICA) was used to separate a multivariate signal into additive subcomponents and ICLabel [30] was applied to remove artifacts caused by muscle activity, heartbeats, eye movements, and eye blinks.

### 2.4. Frontal Alpha Asymmetry (FAA)

It is known that eye closing would alter the amplitude of alpha oscillation and FAA would have high stability representing alpha asymmetry recorded when the patient’s eyes were open [6]. In this study, hf-tRNS was applied in an eyes-open state. Given these considerations, only accepted epochs of eyes-open EEG data collected in a resting state and during stimulation were selected for power spectral analysis using fast Fourier transforms to obtain absolute power in the alpha (8–12.5 Hz) frequency band. The FAA analysis for EEG data was performed using EEBLAB Frontal Alpha Asymmetry Toolbox by subtracting the log-transformed power density values within the alpha frequency range for each homologous left and right frontal electrode pair (F4-F3, Fp2-Fp1, F8-F7) [7]. The FAA can be calculated as follows.
FAA = mean[abs(log(Power_Right) − log(Power_Left))]

Since alpha power is often interpreted as inversely related to cortical activity, higher FAA values reflect greater left-side metabolic frontal brain activity [8,9].

### 2.5. Statistical Analyses

All statistical analyses were performed using IBM SPSS Statistics 21.0 software (IBM SPSS Inc., Chicago, IL, USA). The effects of hf-tRNS on electroencephalographic FAA measures over time were analyzed using repeated measures analysis of variance (RMANOVA) including “time” as the within-group factor (baseline, during the first session of stimulation, the end of stimulation, and the 1-week follow-up visit) and “treatment group” (active versus sham) as the between-group factor. Post-hoc statistical tests were performed using student’s *t*-test across participants, and multiple comparisons were corrected by the false discovery rate (FDR) method. Spearman rank correlations were used to analyze the relationships between the changes in electroencephalographic FAA from baseline to different time points of measurements and treatment response to hf-tRNS. Statistical significance for the results was set at *p* < 0.05 (two-tailed), and the FDR was used for multiple comparisons correction.

## 3. Results

### 3.1. Effects of hf-tRNS on the AA and EXP Domains of Negative Symptoms

There were no between-group significant differences in the scores of FSNS (*p* = 0.62), AA domain (*p* = 0.61), and EXP domain (*p* = 0.87) at baseline. The hf-tRNS group had significantly greater reductions in the scores of FSNS (Figure 2A), AA domain (Figure 2B), and EXP domain (Figure 2C) than the sham condition at the end of stimulation and follow-ups (See Appendix A for details of values and statistics).

### 3.2. Effects of hf-tRNS on Disorganized Symptoms and Extrapyramidal Symptoms

There were no between-group significant differences in the scores of PANSS disorganization dimension (*p* = 0.71) and Extrapyramidal Symptoms Rating Scale (ESRS) (*p* = 0.47) at baseline. The hf-tRNS group had significantly greater reductions in the scores of PANSS disorganization dimension (Figure 3A) and ESRS (Figure 3B) than then sham condition at the end of stimulation and follow-ups (See Appendix A for details of values and statistics).

### 3.3. Effects of hf-tRNS on EEG Frontal Alpha Asymmetry

The absolute power within the alpha frequency range of individual electrodes for calculating FAA (Fp1, F3, F7, Fp2, F4, and F8) was obtained, and the scalp power topography was shown in Figure 4. At the individual electrode level, there were no significant differences in absolute alpha-band power at baseline between the hf-tRNS group and sham group (all *p* values > 0.05). RMANOVA did not show any significant group-by-time interaction for absolute alpha-band power at individual electrode levels (all *p* values > 0.05). There was a significant condition x session interaction effect for FAA found in F4-F3 (F3,31 = 3.71, *p* = 0.022). Post-hoc analyses showed that hf-tRNS group had a greater increase in FAA (F4-F3) during the first stimulation session (t = 2.87, *p* = 0.007) and at the one-week follow-up (t = 3.08, *p* = 0.004) than sham group (Figure 5).

### 3.4. Correlation Analyses

In the hf-tRNS group, FAA at baseline and FAA changes during the first stimulation failed to predict the improvement in the severity of negative symptoms as a whole, and two negative symptom domains (including AA and EXP domain scores) at the end of stimulation and the follow-up visits. There were no statistically significant correlations between changes in FAA and changes in the severity of negative symptoms at all postbaseline visits. All these results remained non-significant when antipsychotic medication dose (in chlorpromazine equivalents) was controlled (See Appendix A for details of the correlations between the doses of antipsychotic medications and EEG-based FAA at baseline).

## 4. Discussion

In this randomized clinical trial including patients with stable patients with schizophrenia who underwent repetitive hf-tRNS over the left prefrontal cortex using a protocol comprising 2 mA, 20 min, twice-daily for a total of 10 sessions with a one-month follow-up, the acute and longer-lasting effects of hf-tRNS on electroencephalographic FAA were investigated. Clinical results for this trial showed that hf-tRNS significantly reduced the severity of negative symptoms at the end of stimulation and the follow-up phase [14]. The major finding of the current EEG study was that hf-tRNS relative to sham increased electroencephalographic FAA during the first stimulation session and at the one-week follow-up. Overall, our study implicated that frontal alpha asymmetry may play a role in the pathophysiological mechanisms underlying negative symptoms of schizophrenia, and EEG-based FAA may be a useful proxy biomarker for a novel treatment under trial in treating negative symptoms.

Negative symptoms were expected to be associated with FAA as they are conceptually related to motivational disturbance, the AA domain of negative symptoms (i.e., motivation- and pleasure-related negative symptoms). However, previous studies failed to detect a correlation between negative symptoms and FAA despite a lower FAA in schizophrenia patients relative to healthy controls [6,7]. Failure to find an association between negative symptoms and FAA in these studies may have been due to either the use of the negative subscale of Positive and Negative Syndrome Scales (PANSS) that measures the negative symptoms as a whole or the use of the Brief Psychiatric Rating Scale (BPRS) negative symptom scale that focuses on the Expressive Deficit (EXP) domain of negative symptoms rather than the Avolition-Apathy (AA) domain of negative symptoms.

As can be seen in Figure 2B, hf-tRNS targeting the left hemispheric lateral prefrontal cortex improved the AA domain of negative symptoms. The left hemispheric lateral PFC is a key node in the frontostriatal networks that serve higher-order motivational and social-drive functions [31]. The impairment in these functions is the core of the AA domain of negative symptoms [32,33]. Research indicates that patients with schizophrenia have impaired lateral prefrontal activity to regulate the intrinsic motivation-action, contributing substantially to their AA domain of negative symptoms [34]. In this trial, the increase in FAA in the hf-tRNS group relative to the sham condition (Figure 4) suggests that hf-tRNS may have increased the left prefrontal activity through its local effect, thereby improving the AA domain of negative symptoms. This is consistent with previous evidence indicating that unidirectional hf-tRNS increases the excitability of the cortex under the anode stimulation site [15]. On the other hand, hf-tRNS significantly improved the EXP domain of negative symptoms (Figure 2C). Previous research investigating the trajectories of two subdomains of negative symptoms indicates that the changes in disorganized symptoms and extrapyramidal symptoms are important factors that affect the improvement of the EXP domain [35]. In this trial, the severity of disorganized symptoms (Figure 3A) and extrapyramidal symptoms (Figure 3B) was significantly reduced in the hf-tRNS group relative to the sham condition. Research indicates that disorganization symptoms of schizophrenia are associated with reduced activity in the left ventrolateral [36] and dorsolateral [37] prefrontal cortex. Increased FAA in the active group relative to sham condition suggests that hf-tRNS may normalize hypofunction in these brain regions and thereby alleviate disorganized symptoms. Moreover, research has reported the potential of activation of the left prefrontal cortex by non-invasive brain stimulation to improve extrapyramidal symptoms in schizophrenia patients [38], possibly through its remote effect on subcortical regions of the dopaminergic system, e.g., increased neural activity in the neostriata system and release of endogenous dopamine in the striatum and putamen [39,40,41]. However, little is known about possible effects at subcortical levels of lateral prefrontal cortex stimulation with unidirectional hf-tRNS. Future studies combining hf-tRNS with functional neuroimaging could clarify the improvements herein observed for both negative symptoms subdomains.

Previous studies are inherently cross-sectional and correlational and cannot establish a causal relationship between FAA and the severity of negative symptoms. In line with an increasing consensus suggesting that researchers experimentally manipulate the hypothesized pathophysiology to clarify the causal relationship between pathophysiological markers and clinical symptoms [42], our study applied hf-tRNS to instantaneously increase FAA (i.e., during the first session of stimulation) in stable patients with schizophrenia and subsequently demonstrated the improvement in the severity of negative symptoms following 10 sessions of stimulation. That is, our results implicated that the asymmetry of frontal activity may be causal to the negative symptoms of schizophrenia. Of note, the study failed to detect a significant increase in FAA at the end of stimulation in the hf-tRNS group relative to the sham condition. The unexpected result may be due to confounding factors during EEG recording (e.g., fatigue, emotional status, and hunger) that were not measured and controlled in this trial [43]. Another possibility is that hf-tRNS over the left lateral prefrontal may not have immediate gains for FAA at the end of the stimulation but have a delayed onset of enhancing effect at a 1-week follow-up.

Our study had limitations. First, the FAA data in the trial may have a potential bias from concomitant medications that may interact with hf-tRNS. However, previous research did not find a significant impact of psychotropic medications on resting frontal EEG asymmetry [44] and our results did not significantly alter after the CPZ equivalents were controlled. Second, we collected EEG data during stimulation and used mathematical methods to remove pronounced stimulation artifacts, allowing an online assessment of sub-threshold stimulation-induced brain alterations and gaining a better understanding of the underlying neurophysiological effects of hf-tRNS. However, it remains technically challenging to have complete and successful removal of stimulation artifacts without distortion of the signals of interest [45,46]. Third, our trial recruited patients with stable schizophrenia. We cannot completely exclude the possibility that the improvement in the overall severity of negative symptoms came from the reduction in secondary negative symptoms, which are hard to be distinguished from primary enduring negative symptoms without explicit information. Future studies in a sample of schizophrenia patients with predominantly negative symptoms are required to confirm whether our electroencephalographic FAA results can be replicated by others. Finally, correlation coefficients of neither the baseline FAA nor the change in FAA on treatment response reached statistical significance. It is possible that the remote effect of hf-tRNS on subcortical neuronal activity rather than its local effect on cortical activity directly correlates with the treatment response of hf-tRNS. But we cannot exclude the possibility of a significant non-linear correlation between the two variables not detected by spearman correlation analyses. Replication in a larger sample will be a practical way to allow a more precise estimate of FAA as a surrogate endpoint for treatment response.

## 5. Conclusions

In summary, the current findings allow a better understanding of the potential neural mechanism underlying the unidirectional hf-tRNS targeting the left lateral prefrontal cortex in treating negative symptoms of schizophrenia. These findings suggest that modulating electroencephalographic frontal alpha asymmetry by using unidirectional hf-tRNS may play a key role in reducing negative symptoms in patients with schizophrenia. In this respect, our results open the possibility for the application of state-of-the-art sub-threshold non-invasive brain stimulation for targeting negative symptoms.

## Figures and Tables

**Figure 1 jpm-12-01667-f001:**
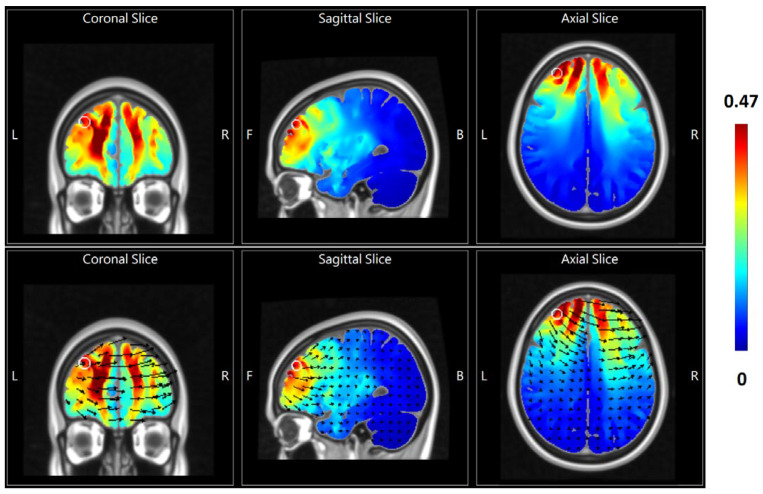
2D representation of electric field simulation of unidirectional high-frequency transcranial random noise stimulation (hf-tRNS) over the left lateral prefrontal cortex by HD-Explore^®^ (Soterix Medical, New York, NY, USA), which utilizes a finite element model of brain current flow based on an MRI derived MNI-152 standard brain template. Colors represent the electric field intensity (V/m). The upper and lower panel represents the simulated electrical field distribution without and with vectors (black arrows) of electrical current flow, respectively. The center of the white circle represents the MNI152 stereotaxic coordinate (x–33, y + 48, z + 33) of the projection of the 10–10 system scalp electrode position AF3 (anodal location of hf-tRNS) onto the cortical surface, where the field intensity was 0.422 V/m.

**Figure 2 jpm-12-01667-f002:**
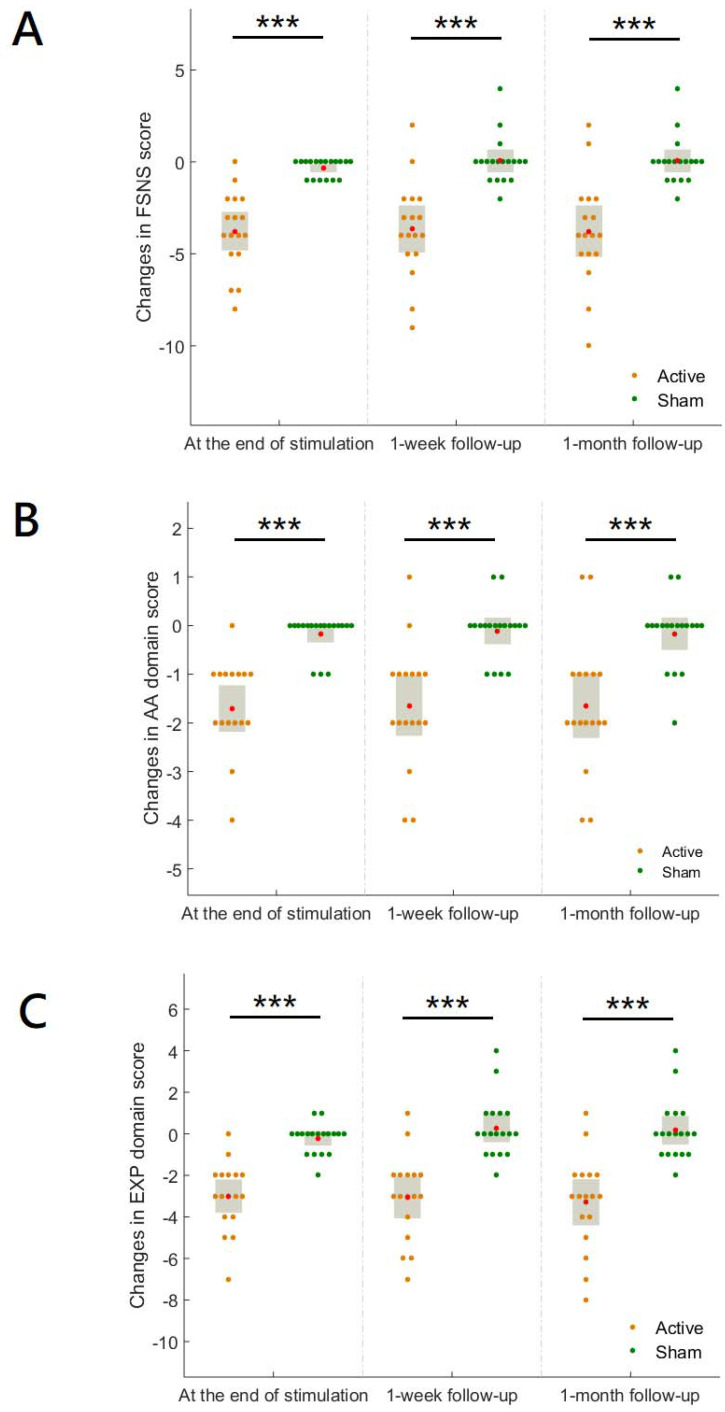
The changes in the scores of Factor Score for Negative Symptoms (FSNS) of the Positive and Negative Syndrome Scale (**A**), the Avolition-Apathy (AA) domain of negative symptoms (**B**), and the Expressive Deficit (EXP) domain of negative symptoms (**C**) at the end of stimulation and follow-ups between the hf-tRNS group and sham condition. The red dot indicated the mean of the data. The gray box indicated a 95% confidence interval. ***, *p* < 0.001.

**Figure 3 jpm-12-01667-f003:**
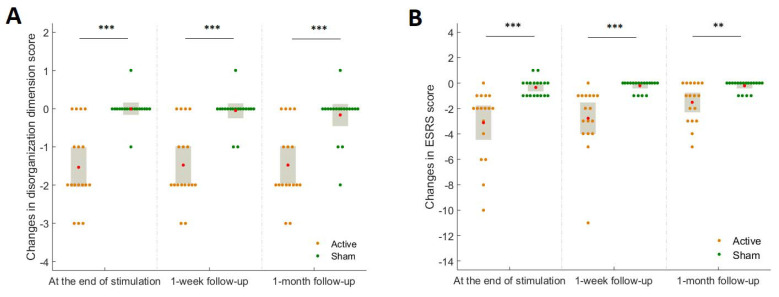
The changes in the scores of PANSS disorganization dimension (**A**) and Extrapyramidal Symptoms Rating Scale (ESRS) (**B**) at the end of stimulation and follow-ups between the hf-tRNS group and sham condition. The red dot indicated the mean of the data. The gray box indicates a 95% confidence interval. **, *p* <0.01, ***, *p* < 0.001.

**Figure 4 jpm-12-01667-f004:**
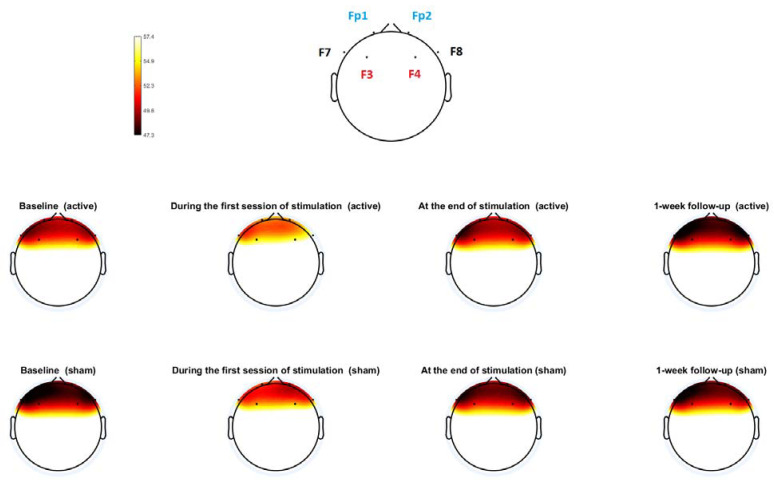
Topoplots of alpha power in the individual electrode for calculating frontal alpha asymmetry (Fp1, F3, F7, Fp2, F4, and F8). There were no significant differences in absolute alpha-band power at baseline between the hf-tRNS group and sham group (all *p* values > 0.05). The changes in the absolute alpha-band power from baseline to the first session of stimulation, the end of stimulation, and 1-week follow-up were not significantly different between the hf-tRNS group and sham condition (all *p* values > 0.05). Colors represent power levels.

**Figure 5 jpm-12-01667-f005:**
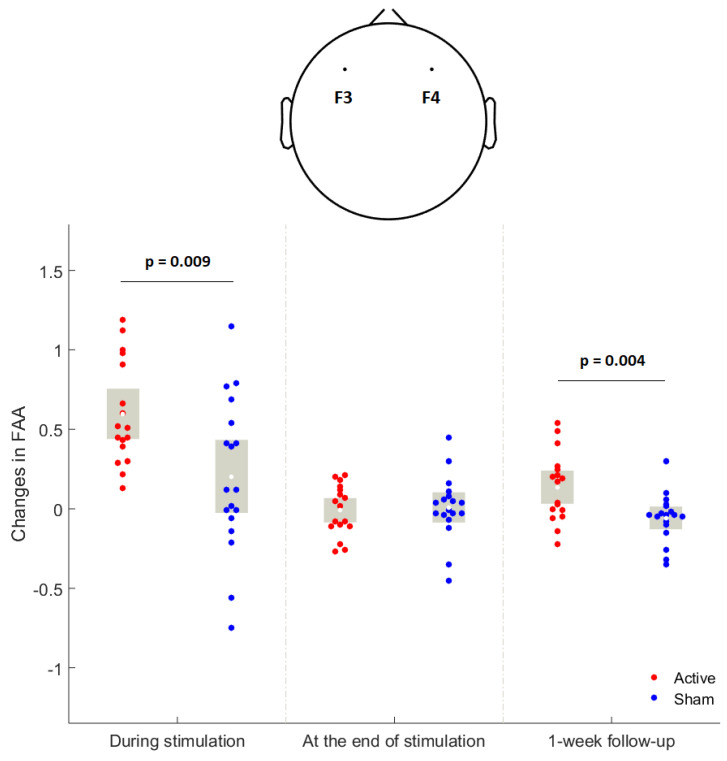
The changes in the electroencephalographic F4-F3 frontal alpha asymmetry (FAA) during stimulation (the first stimulation session), at the end of stimulation, and 1-week follow-up between the hf-tRNS group and sham condition. The white dot indicates the mean of the data. The gray box indicated a 95% confidence interval.

**Table 1 jpm-12-01667-t001:** Concise demographics and clinical data of the participants.

	hf-tRNS (N = 17)	Sham (N = 18)	*p* Value
Schizophrenia/schizoaffective disorder	12/5	13/5	0.92
Gender (f/m)	6/11	8/10	0.58
Handedness (r/l)	15/2	16/2	1.00
Age, years old	44.06 ± 12.50	43.17 ± 11.63	0.76
Years of education, years	13.53 ± 2.32	12.44 ± 3.52	0.42
Years since diagnosis, years	18.82 ± 9.73	19.11 ± 13.35	0.94
Chlorpromazine equivalent dose, mg/day	581.70 ± 310.59	626.10 ± 298.82	0.67
PANSS total score	69.00 ± 9.64	71.39 ± 9.06	0.46
PANSS Factor Score for Negative Symptoms (FSNS)	21.29 ± 3.00	21.94 ± 4.40	0.62
AA domain of negative symptoms	9.06 ± 1.20	9.33 ± 1.85	0.69
EXP domain of negative symptoms	16.94 ± 2.38	17.11 ± 3.50	0.87
PANSS Factor Score for Positive Symptoms	11.71 ± 3.46	12.61 ± 3.74	0.46
PANSS Factor Score for Excitement	5.59 ± 2.48	5.28 ± 1.81	0.78
PANSS Factor Score for Disorganization	11.88 ± 1.27	12.11 ± 2.14	0.71
PANSS Factor Score for Emotional distress	5.82 ± 2.46	6.39 ± 1.50	0.10
ESRS score	10.12 ± 10.17	7.94 ± 7.08	0.13

Abbreviations: hf-tRNS, High-frequency transcranial random noise stimulation; PANSS, Positive and Negative Syndrome Scale; FSNS, Factor Score for Negative Symptoms; AA, the Avolition-Apathy; EXP, the Expressive Deficit; ESRS, Extrapyramidal Symptoms Rating Scale. Notes: Data are presented as means ± standard deviations unless otherwise stated.

## Data Availability

The data presented in this study are available on request from the corresponding author.

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
