# Peer review of "High-Frequency Transcranial Random Noise Stimulation over the Left Prefrontal Cortex Increases Resting-State EEG Frontal Alpha Asymmetry in Patients with Schizophrenia"

_jpm, 2022, doi:10.3390/jpm12101667_

Round 1

Reviewer 1 Report

Interesting and well-written work. Iwould like to know whether The Authors Recorded ieeg at the follow up visit, if they computeded some Network topholohy Analysis and wether they assessed the Effects of pharmachological therapy on the eeg findings. Lastly, how are these findings relevant for the clinical practice?

Author Response

Reviewer 1

Interesting and well-written work. Iwould like to know whether The Authors Recorded ieeg at the follow up visit, if they computeded some Network topholohy Analysis and wether they assessed the Effects of pharmachological therapy on the eeg findings. Lastly, how are these findings relevant for the clinical practice?

Comment 1

whether The Authors Recorded ieeg at the follow up visit,

Response 1

We would like to thank the reviewer for the time spent and the precious comments. Our study has recorded EEG at baseline, during the first session of stimulation, at the end of stimulation, and one-week follow-up. At one-month follow-up, we did not record EEG.

Comment 2

if they computeded some Network topholohy Analysis

Response 2

We have performed EEG source-level network analyses at gamma-frequency band and the results will be published in this special issue of journal of personalized medicine. Please see the manuscript entitled “ High-Frequency Transcranial Random Noise Stimulation Modulates Gamma-Band EEG Source-Based Large-Scale Functional Network Connectivity in Patients with Schizophrenia: A Randomized, Double-Blind, Sham-Controlled Clinical Trial”  and being published in J. Pers. Med.  2022, 12,1617. https://doi.org/10.3390/jpm12101617. However, the results of EEG scalp-level and source-level network analyses at other frequency bands were all negative.

Comment 3

the Effects of pharmachological therapy on the eeg findings.

Response 3

As suggested by the reviewer, we add correlation analyses in Table S2.

Comment 4

Lastly, how are these findings relevant for the clinical practice?

Response 4

As suggested by the reviewer, we revise the discussion in line 323-326. 

Reviewer 2 Report

I thank the Authors and Editor for the opportunity to review this interesting study. In my opinion this study is very relevant  because, in addition to ascertaining the effectiveness of tRNS in improving negative symptoms in patients with schizophrenia, it offers further insights regarding the possible relationships between the improvements of other psychopathological domains and the effects of NIBS. In particular, the role of NIBS to positively influence disorganized symptoms appears increasingly clear.

Nevertheless, several points deserve further attention before publication.

Introduction

Line 86: the reference of Tang et al., 2022 is very punctual addressing the efficacy of HF-tRNS in the treatment of negative symptoms. 

To enrich the quality of the manuscript I suggest adding a more recent study by Lisoni et al. 2022 that demonstrated negative symptoms improvements by prefrontal-tDCS, particularly in the domains of Avolition-Apathy and Expressive Deficit. This is a relevant matter since the Authors considered the effects of tRNS on both these dimensions. This is an important distinction as NIBS are supposed to differently act on these dimensions as different brain networks seem to be, probably, involved for each dimension of negative symptoms, as coherently reported by the Authors in the Introduction.

Please see,

Lisoni, J., Baldacci, G., Nibbio, G., Zucchetti, A., Lemmi Gigli, E.B., Savorelli, A., Facchi, M., Miotto, P., Deste, G., Barlati, S., Vita, A., 2022. Effects of bilateral, bipolar-nonbalanced, frontal transcranial Direct Current Stimulation (tDCS) on negative symptoms and neurocognition in a sample of patients living with schizophrenia: Results of a randomized double-blind sham-controlled trial. J. Psychiatr. Res. https://doi.org/https://doi.org/10.1016/j.jpsychires.2022.09.011

In this context, I suggest to modify some definitions in the section 2.1 of Materials and Methods.  Indeed, the Authors considered “Two subdomains of negative symptoms were identified from the PANSS: Expressive Negative Symptoms (Exp Neg, the sum of items of N1, N3, N6, G5, G7 and G13) and Social Amotivation (Soc Amot, the sum of items of N2, N4, and 122 G16), i.e., the experiential negative symptoms”. 

The terms “Expressive Negative Symptoms” and “Social Amotivation” are very confusing and scientifically misleading. I think that the Authors used these terms as in Cella et al. 2018. As these definitions are unclear and unrecognized at international level, I suggest to refer to Liemburg et al. 2013 and Galderisi et al., 2018 where the words to define each dimensions of negative symptoms are quite different in term of language. 

Moreover, as Cella et al., 2018 linked to Liemburg et al., 2013 definition, the Authors applied the factorial structure primarily suggested by Liemburg et a., 2013, not by Cella et al., 2018. 

Galderisi et al. 2018 defined the Avolition–apathy domain and the Expressive deficit domain in a very precise way: these definitions are in line with current classification of negative symptoms in schizophrenia that are multidimensionally described as two distinct factorsby international expert in the field (Kirkpatrick et al., 2006):

·      the Expressive Deficit domain (EXP domain) including features of blunted affect and alogia

·      the Avolition-Apathy domain(AA domain) including manifestations of anhedonia, asociality andavolition) 

Further explanations on neural and clinical differences among Avolition–apathy domain and Expressive deficit domain could be found in Begue et al., 2020. 

Thus, in order to avoid possible misinterpretation at methodological level and achieving international recognized definitions, I suggest to apply the definition of negative symptoms domains according to Galderisi et al., 2018, changing the term of “Expressive Negative Symptoms” with  “Expressive deficit domain” and “Social Amotivation” with “Avolition–apathy domain” throughout all the manuscript, including tables and figures.

 Please see,

Galderisi, S., Mucci, A., Buchanan, R.W., Arango, C., 2018. Negative symptoms of schizophrenia: new developments and unanswered research questions. The Lancet Psychiatry 5, 664–677. https://doi.org/10.1016/S2215-0366(18)30050-6

Liemburg, E., Castelein, S., Stewart, R., van der Gaag, M., Aleman, A., Knegtering, H., Kahn, R.S., Linszen, D.H., van Os, J., Wiersma, D., Bruggeman, R., Cahn, W., de Haan, L., Krabbendam, L., Myin-Germeys, I., 2013. Two subdomains of negative symptoms in psychotic disorders: Established and confirmed in two large cohorts. J. Psychiatr. Res. 47, 718–725. https://doi.org/10.1016/j.jpsychires.2013.01.024

Bègue, I., Kaiser, S., Kirschner, M., 2020. Pathophysiology of negative symptom dimensions of schizophrenia – Current developments and implications for treatment. Neurosci. Biobehav. Rev. 116, 74–88. https://doi.org/https://doi.org/10.1016/j.neubiorev.2020.06.004

Kirkpatrick, B., Fenton, W.S., Carpenter  Jr, W.T., Marder, S.R., 2006. The NIMH-MATRICS consensus statement on negative symptoms. Schizophr. Bull. 32, 214–219. https://doi.org/10.1093/schbul/sbj053

Materials and Methods

Line 120: the Authors considered the PANSS Factor structure derived from Marder et al., 1997 to analyze negative symptoms. Did the Authors also apply the same factorialization for the other psychopathological factors? Please clarify. 

Moreover, no mention has been provided in Material and Methods regarding the Extrapyramidal Symptoms Rating Scale (ESRS), although some results is provided regarding the reduction of ESRS scores. Please add some details in the Material and Methods regarding the assessment with ESRS.

I recommend a general revision of the assessment tools, also considering the timepoint in which clinical assessment is conducted, as at the moment it is quite confusing for the reader.

Line 136: the Authors considered their electrode montage as being a high-definition 4 × 1 electrode montage. However, the definition of “high definition” is somehow confusing since HD montage generally refers to conditions in which the anodic electrode is surrounded by 4 cathodes. This is not the case of the present study where the anode was placed at AF3 and cathodes over AF4, F2, F6, and FC4. Moreover, cathodes are placed over the right hemisphere while the anode is on the left one. It is recommended to provide a punctual explanation on the Authors’ decision to apply this electrode montage. Did the Authors consider previous combined NIBS-neuroimaging results? 

Line 137. It is welcomed to add information of electrodes’ dimensions.

Line 140-145: the Authors reported that “Active hf-tRNS delivered twice-daily, 20-min, and 2 140 mA-intensity random noise stimulation with 100-640 Hz frequency, 1 mA offset, and 15 s 141 ramp-in/ramp-out for 5 consecutive weekdays.” 

Here, the duration of stimulation protocol is unprecise. In the abstract is written that stimulation protocol consisted in five-day, twice-daily hf-tRNS. Please uniform the like this “ten session of … were performed twice-a-day for 5 consecutive days. Form Monday to Friday?

Results

Table 1: please add statistic values for each considered variable.

I suggest describing in a separate section the baseline characteristics of the sample, reporting possible differences of clinical variables, avoiding to isolate these results in multiple paragraphs. 

Figures are very interesting and well designed to express the results of the present study. However, I suggest to build a separate table regarding the improvements of negative symptoms, Expressive Negative Symptoms, Avolition–apathydomain, disorganization factor and ESRS score achieved by hf-tRNS with appropriate statistical analyses. 

Discussion

The argumentation is very well written and scientifically relevant, especially when the Authors consider possible relation between disorganization symptoms/extrapyramidal symptoms and expressive domain impairments. 

However, as the Authors found improvements in both negative symptoms subdomains as in Lisoni et al., 2022, I suggest to revise lines 230-232 with the idea that NIBS interventions could exert local effects at cortical level but little is known on possible effects at subcortical levels. One suggestion could be that combining NIBS with functional neuroimaging could clarify the improvements herein observed for both negative symptoms subdomains. Similarly, the Authors found that FAA at baseline and FAA changes during the first stimulation failed to predict the improvement in the severity of negative symptoms as a whole and two negative symptom subdomains. This is due to focal effects due to tRNS? A discussion on this topic is welcome by the Authors, also considering possible implication of tRNS on subcortical structures. 

Author Response

Reviewer 2

I thank the Authors and Editor for the opportunity to review this interesting study. In my opinion this study is very relevant  because, in addition to ascertaining the effectiveness of tRNS in improving negative symptoms in patients with schizophrenia, it offers further insights regarding the possible relationships between the improvements of other psychopathological domains and the effects of NIBS. In particular, the role of NIBS to positively influence disorganized symptoms appears increasingly clear.

Nevertheless, several points deserve further attention before publication.

Comment 1

Introduction

Line 86: the reference of Tang et al., 2022 is very punctual addressing the efficacy of HF-tRNS in the treatment of negative symptoms. 

To enrich the quality of the manuscript I suggest adding a more recent study by Lisoni et al. 2022 that demonstrated negative symptoms improvements by prefrontal-tDCS, particularly in the domains of Avolition-Apathy and Expressive Deficit. This is a relevant matter since the Authors considered the effects of tRNS on both these dimensions. This is an important distinction as NIBS are supposed to differently act on these dimensions as different brain networks seem to be, probably, involved for each dimension of negative symptoms, as coherently reported by the Authors in the Introduction.

Please see,

Lisoni, J., Baldacci, G., Nibbio, G., Zucchetti, A., Lemmi Gigli, E.B., Savorelli, A., Facchi, M., Miotto, P., Deste, G., Barlati, S., Vita, A., 2022. Effects of bilateral, bipolar-nonbalanced, frontal transcranial Direct Current Stimulation (tDCS) on negative symptoms and neurocognition in a sample of patients living with schizophrenia: Results of a randomized double-blind sham-controlled trial. J. Psychiatr. Res. https://doi.org/https://doi.org/10.1016/j.jpsychires.2022.09.011

 Response 1

As suggested by the insightful comment from the reviewer, we add a new reference no.13.

In this context, I suggest to modify some definitions in the section 2.1 of Materials and Methods.  Indeed, the Authors considered “Two subdomains of negative symptoms were identified from the PANSS: Expressive Negative Symptoms (Exp Neg, the sum of items of N1, N3, N6, G5, G7 and G13) and Social Amotivation (Soc Amot, the sum of items of N2, N4, and 122 G16), i.e., the experiential negative symptoms”. 

The terms “Expressive Negative Symptoms” and “Social Amotivation” are very confusing and scientifically misleading. I think that the Authors used these terms as in Cella et al. 2018. As these definitions are unclear and unrecognized at international level, I suggest to refer to Liemburg et al. 2013 and Galderisi et al., 2018 where the words to define each dimensions of negative symptoms are quite different in term of language. 

Moreover, as Cella et al., 2018 linked to Liemburg et al., 2013 definition, the Authors applied the factorial structure primarily suggested by Liemburg et a., 2013, not by Cella et al., 2018. 

Galderisi et al. 2018 defined the Avolition–apathy domain and the Expressive deficit domain in a very precise way: these definitions are in line with current classification of negative symptoms in schizophrenia that are multidimensionally described as two distinct factorsby international expert in the field (Kirkpatrick et al., 2006):

  • the Expressive Deficit domain (EXP domain) including features of blunted affect and alogia
  • the Avolition-Apathy domain(AA domain) including manifestations of anhedonia, asociality andavolition) 

Further explanations on neural and clinical differences among Avolition–apathy domain and Expressive deficit domain could be found in Begue et al., 2020. 

Thus, in order to avoid possible misinterpretation at methodological level and achieving international recognized definitions, I suggest to apply the definition of negative symptoms domains according to Galderisi et al., 2018, changing the term of “Expressive Negative Symptoms” with  “Expressive deficit domain” and “Social Amotivation” with “Avolition–apathy domain” throughout all the manuscript, including tables and figures.

 Please see,

Galderisi, S., Mucci, A., Buchanan, R.W., Arango, C., 2018. Negative symptoms of schizophrenia: new developments and unanswered research questions. The Lancet Psychiatry 5, 664–677. https://doi.org/10.1016/S2215-0366(18)30050-6

Liemburg, E., Castelein, S., Stewart, R., van der Gaag, M., Aleman, A., Knegtering, H., Kahn, R.S., Linszen, D.H., van Os, J., Wiersma, D., Bruggeman, R., Cahn, W., de Haan, L., Krabbendam, L., Myin-Germeys, I., 2013. Two subdomains of negative symptoms in psychotic disorders: Established and confirmed in two large cohorts. J. Psychiatr. Res. 47, 718–725. https://doi.org/10.1016/j.jpsychires.2013.01.024

Bègue, I., Kaiser, S., Kirschner, M., 2020. Pathophysiology of negative symptom dimensions of schizophrenia – Current developments and implications for treatment. Neurosci. Biobehav. Rev. 116, 74–88. https://doi.org/https://doi.org/10.1016/j.neubiorev.2020.06.004

Kirkpatrick, B., Fenton, W.S., Carpenter  Jr, W.T., Marder, S.R., 2006. The NIMH-MATRICS consensus statement on negative symptoms. Schizophr. Bull. 32, 214–219. https://doi.org/10.1093/schbul/sbj053

Response 2 

As suggested by the insightful comment from the reviewer, we change the term of “Expressive Negative Symptoms” with  “Expressive deficit domain” and “Social Amotivation” with “Avolition–apathy domain” throughout all the manuscript, including tables and figures. We also add the new references as suggested.

Comment 3

Materials and Methods

Line 120: the Authors considered the PANSS Factor structure derived from Marder et al., 1997 to analyze negative symptoms. Did the Authors also apply the same factorialization for the other psychopathological factors? Please clarify. 

Response 3

As suggested by the insightful comment from the reviewer, we revise our methods in line 127-129.

Comment 4

Moreover, no mention has been provided in Material and Methods regarding the Extrapyramidal Symptoms Rating Scale (ESRS), although some results is provided regarding the reduction of ESRS scores. Please add some details in the Material and Methods regarding the assessment with ESRS.

Response 4

As suggested by the insightful comment from the reviewer, we revise our methods in line 129-134.

Comment 5

I recommend a general revision of the assessment tools, also considering the timepoint in which clinical assessment is conducted, as at the moment it is quite confusing for the reader.

Response 5

As suggested by the insightful comment from the reviewer, we add a figure S1 in the supplementary material.

Comment 6

Line 136: the Authors considered their electrode montage as being a high-definition 4 × 1 electrode montage. However, the definition of “high definition” is somehow confusing since HD montage generally refers to conditions in which the anodic electrode is surrounded by 4 cathodes. This is not the case of the present study where the anode was placed at AF3 and cathodes over AF4, F2, F6, and FC4. Moreover, cathodes are placed over the right hemisphere while the anode is on the left one. It is recommended to provide a punctual explanation on the Authors’ decision to apply this electrode montage. Did the Authors consider previous combined NIBS-neuroimaging results? 

Response 6

 As suggested by the insightful comment from the reviewer, we change “high definition” to “high-density”  in line 148 and revise our methods in line 153-157

References

Chan HN, Alonzo A, Martin DM, Player M, Mitchell PB, Sachdev P, Loo CK. Treatment of major depressive disorder by transcranial random noise stimulation: case report of a novel treatment. Biol Psychiatry. 2012 Aug 15;72(4):e9-e10. doi: 10.1016/j.biopsych.2012.02.009. Epub 2012 Mar 17. PMID: 22425412.

Helfrich RF, Knepper H, Nolte G, Strüber D, Rach S, Herrmann CS, Schneider TR, Engel AK. Selective modulation of interhemispheric functional connectivity by HD-tACS shapes perception. PLoS Biol. 2014 Dec 30;12(12):e1002031. doi: 10.1371/journal.pbio.1002031. PMID: 25549264; PMCID: PMC4280108.

Comment 7

Line 137. It is welcomed to add information of electrodes’ dimensions.

Response 7

 As suggested by the insightful comment from the reviewer, we add information of electrodes’ dimensions in line 149-150

Comment  8

Line 140-145: the Authors reported that “Active hf-tRNS delivered twice-daily, 20-min, and 2  mA-intensity random noise stimulation with 100-640 Hz frequency, 1 mA offset, and 15 s ramp-in/ramp-out for 5 consecutive weekdays.” 

Here, the duration of stimulation protocol is unprecise. In the abstract is written that stimulation protocol consisted in five-day, twice-daily hf-tRNS. Please uniform the like this “ten session of … were performed twice-a-day for 5 consecutive days. Form Monday to Friday?

Response 8

 As suggested by the insightful comment from the reviewer, we revise the abstract in line 30 and our methods in line 155-157

Comment  9

Results

Table 1: please add statistic values for each considered variable.

I suggest describing in a separate section the baseline characteristics of the sample, reporting possible differences of clinical variables, avoiding to isolate these results in multiple paragraphs. 

Response 9

  As suggested by the insightful comment from the reviewer, we add statistic values in the Table 1.

Comment  10

Figures are very interesting and well designed to express the results of the present study. However, I suggest to build a separate table regarding the improvements of negative symptoms, Expressive Negative Symptoms, Avolition–apathydomain, disorganization factor and ESRS score achieved by hf-tRNS with appropriate statistical analyses. 

 Response  10

  As suggested by the insightful comment from the reviewer, we add Table S1 in the supplementary material.

Comment  11

Discussion

The argumentation is very well written and scientifically relevant, especially when the Authors consider possible relation between disorganization symptoms/extrapyramidal symptoms and expressive domain impairments. 

However, as the Authors found improvements in both negative symptoms subdomains as in Lisoni et al., 2022, I suggest to revise lines 230-232 with the idea that NIBS interventions could exert local effects at cortical level but little is known on possible effects at subcortical levels. One suggestion could be that combining NIBS with functional neuroimaging could clarify the improvements herein observed for both negative symptoms subdomains. Similarly, the Authors found that FAA at baseline and FAA changes during the first stimulation failed to predict the improvement in the severity of negative symptoms as a whole and two negative symptom subdomains. This is due to focal effects due to tRNS? A discussion on this topic is welcome by the Authors, also considering possible implication of tRNS on subcortical structures. 

 Response  11

  As suggested by the insightful comment from the reviewer, we revise our discussion in line 365-369 and in line 401-404.

Round 2

Reviewer 2 Report

The Authors substantially reviewed the manuscript according to suggestion.

Please add reference 13 as follows as in the present form it is incorrect:

Lisoni, J., Baldacci, G., Nibbio, G., Zucchetti, A., Lemmi Gigli, E.B., Savorelli, A., Facchi, M., Miotto, P., Deste, G., Barlati, S., Vita, A., 2022. Effects of bilateral, bipolar-nonbalanced, frontal transcranial Direct Current Stimulation (tDCS) on negative symptoms and neurocognition in a sample of patients living with schizophrenia: Results of a randomized double-blind sham-controlled trial. J. Psychiatr. Res. https://doi.org/https://doi.org/10.1016/j.jpsychires.2022.09.011